# Idiopathic Pulmonary Arterial Hypertension in Paediatrics Represents Still a Serious Challenge: A Case Series Study

**DOI:** 10.3390/children10030518

**Published:** 2023-03-06

**Authors:** Pier Paolo Bassareo, Paola Argiento, Colin Joseph McMahon, Esme Dunne, Kevin Patrick Walsh, Maria Giovanna Russo, Michele D’Alto

**Affiliations:** 1Mater Misercordiae University Hospital, Mater, D07 R2WY Dublin, Ireland; 2Children’s Health Ireland at Crumlin, D12 N512 Dublin, Ireland; 3School of Medicine, University College Dublin, Belfield, D04 V1W8 Dublin, Ireland; 4UOC Cardiologica Pediatrica, Dipartimento di Cardiologia, Università della Campania “Luigi Vanvitelli”, Ospedale Monaldi, AORN dei Colli, 80131 Naples, Italy; 5Pulmonary Hypertension Unit, Dipartimento di Cardiologia, Università della Campania “Luigi Vanvitelli”, Ospedale Monaldi, AORN dei Colli, 80131 Naples, Italy

**Keywords:** idiopathic pulmonary arterial hypertension, children, adolescents, echocardiography, right heart catheterisation, lung transplantation

## Abstract

Introduction: Paediatric pulmonary hypertension (PH) represents a heterogeneous illness that is responsible for high morbidity and mortality if left without treatment. Idiopathic pulmonary arterial hypertension (IPAH) is a subtype of PAH rarely seen in paediatrics. Limited long-term data are available. Methods: Over a period of 20 years, 10 paediatric patients were enrolled at two tertiary centres. Their clinical, echocardiographic, and right heart catheterisation (RHC) features and outcome were evaluated. Results: The mean age at first diagnosis was 5.7 ± 5.7 years. The age at the last follow-up was 12.4 ± 6.1 years. The average follow-up was 6.6 ± 0.8 years. There was a female prevalence of 60% (*p* < 0.05) in this case series. Regarding the NYHA functional class, 80% of IPAH subjects were in class III or IV. The mean saturation was 91 ± 5%. In this regard, 70% of the patients were on a combination of three drugs, with sildenafil (90%) included. On echocardiography, longitudinal right ventricular contractility (TAPSE) was slightly reduced (13.4 ± 2.6 mm), whilst RVSP was severely elevated (101 ± 19 mmHg). The RHC data showed that mPAP was 61.8 ± 23.1 mmHg (*p* = 0.0017 with RVSP on echocardiography), mRAP was 10.7 ± 3.8 mmHg, CI was 2.6 ± 1 L·min^−1^·m^−2^, PVRi was 16.8 ± 12.6 WU·m^2^, and SVO_2_ was 63.6 ± 14.8%. Regarding the outcome, two male IPAH patients (20%) died, and 50% underwent lung transplant or were on transplant assessment or already on the waiting list for lung transplantation. One patient underwent a ductus arteriosus stenting (reverse Potts shunt) and another underwent atrial septostomy and stenting. Conclusions: Notwithstanding the progress in medical therapy, IPAH continues to represent a serious challenge, particularly in the paediatric population, with the need for lung transplantation and significant mortality.

## 1. Introduction

Paediatric pulmonary hypertension (PH) represents a heterogeneous illness that is responsible for high morbidity and mortality if left without treatment [1]. The definition of PH in paediatrics is the same as that used in the adult setting. In fact it is represented by the harmful combination of increased mean pulmonary artery pressure (mPAP) at rest >20 mmHg, pulmonary arterial wedge pressure (PAWP) ≤15 mmHg, and pulmonary vascular resistance index (PVRi) >3 WU·m^2^ in patients with biventricular physiology being subjected to right heart catheterisation (RHC) [2]. Systemic blood pressure in paediatrics shows fluctuations related to age and height. Hence, also pulmonary-to-systemic pressure ratio >0.4 can suggest the presence of PAH in children [3].

Idiopathic pulmonary arterial hypertension (IPAH) is a subtype of PAH rarely seen in paediatrics. An exact underlying risk factor is unknown. It is classified in group I PH, together with PH triggered by congenital heart disease [4]. IPAH can manifest in newborns or infants in the form of persistent pulmonary hypertension of the newborn (PPHN) [5]. In these patients, lung histopathology changes are very similar to those seen in older patients suffering from IPAH [6].

However, little is known on outcomes of the disease in paediatrics. The aim of this case series is to evaluate the clinical, echocardiographic, and RHC characteristics and outcome in a small group of IPAH children and adolescents who were followed up at two tertiary centres.

## 2. Materials and Methods

Over a period of 20 years (2002–2022), IPAH was diagnosed in 10 children and adolescents < 18 years old according to the criteria of the 2018 World Symposium on PH, which have tried to conform the PH codification in paediatrics to that used in the adult setting [7].

The exclusion criteria were children with any other aetiologies of PH other than IPAH and people aged over 18 years.

Clinical, echocardiographic, and RHC characteristics and outcome of the enrolled subjects were analysed.

The study was conducted in accordance with the Declaration of Helsinki. Ethics Committee approval was waived since this is a retrospective audit with a small number of patients.

### 2.1. Echocardiography

The scan was carried out in the left lateral decubitus position with a Philips iE33 ultrasound machine (Koninklijke Philips N.V., Amsterdam, The Netherlands), using a 12–5 MHz probe, in accordance with international recommendations [8]. The right ventricular longitudinal systolic function was recorded in terms of tricuspid annular plane systolic excursion (TAPSE). The right ventricular systolic pressure (RVSP) was calculated from tricuspid valve insufficiency peak velocity (modified Bernoulli equation) plus the estimated right atrial pressure.

Acquisition of the above-stated parameters was from an apical view. Six to eight beats were gathered for evaluation keeping away from images of pre- and post-extrasystolic beats. At least three consecutive measurements were taken on collected images and the average was used for analysis.

### 2.2. Right Heart Catheterisation

All the patients underwent RHC to measure the right-sided cardiac pressures and vascular resistances as well as to calculate cardiac output invasively [9].

A local anaesthetic was provided subcutaneously at the access site in the femoral, antecubital, or jugular areas. Venous access was obtained through anatomical landmarks or with echographic guidance. Subsequently, an appropriate sheath was inserted into the vein and fixed securely. A pulmonary artery wire was placed through the sheath into the vein. The balloon at the tip of the wire was inflated for an easier advancement to the right cardiac chambers. The wire advancement was detected by fluoroscopy. When it reached the right atrium, the right atrial pressure (mRAP mmHg) was recorded. Afterward, the wire was advanced into the right ventricle, and intraventricular pressure was recorded. The wire was moved ahead to measure the pulmonary capillary wedge pressure. After that, the balloon was deflated and then brought back a few centimetres into the pulmonary artery to measure mean pulmonary artery pressure (mPAP mmHg). All cardiac pressures were collected at end-expiration averaging at least three cardiac cycles. Blood samples were withdrawn in superior vena cava, inferior vena cava, right atrium, right ventricle, and main pulmonary artery (mixed venous oxygen saturation—SVO_2_) to rule out intracardiac shunts and calculate cardiac output with the Fick principle. Thermodilution was repeated at least three times to obtain an average cardiac output and cardiac index (CI L·min^−1^ ·m^−2^, e.g., the cardiac output indexed for body surface area) [10,11].

Finally, the pulmonary vascular resistance index (PVRi WU∗m^2^) was calculated through the formula: [(Mean pulmonary artery pressure−Pulmonary capillary wedge pressure)/Cardiac index] [11].

## 3. Statistics

All data are presented as percentage or mean ± standard deviation. The Chi-squared and Mann–Whitney U tests were used to make a comparison between genders. Statistical significance was set at *p* < 0.05 throughout the study.

## 4. Results

IPAH patients with their features are summarised in Appendix A. Echocardiography and RHC are performed at the last follow-up.

The age range at first diagnosis was between 1 month and 17 years (mean 5.7 ± 5.7 years). Two children (20%) were diagnosed before 1 year of age. The age at the last follow-up was 12.4 ± 6.1 years. The average follow-up was 6.6 ± 0.8 years. There was a female prevalence of 60% in the case series. Regarding the NYHA functional class, most of the IPAH subjects were in class III (40%) or IV (40%). The mean saturation on room air was 91 ± 5%. As to therapy, 70% of the patients were on a combination of three drugs, with sildenafil (90%) included. On echocardiography, right ventricular contractility, expressed as TAPSE, was slightly reduced (13.4 ± 2.6 mm), whilst RVSP was severely elevated (101 ± 19 mmHg). The RHC data showed that mPAP was 61.8 ± 23.1 mmHg, mRAP was 10.7 ± 3.8 mmHg, CI was 2.6 ± 1 L·min^−1^·m^−2^, PVRi was 16.8 ± 12.6 WU·m^2^, and SVO_2_ 63.6 ± 14.8%. The estimated RVSP by Doppler was significantly different compared to right ventricular pressure that was measured on RHC (*p* = 0.0017). Regarding the outcome, two IPAH patients (20%) died, and 50% underwent lung transplant or were on transplant assessment or already on the waiting list for lung transplantation. One patient underwent a ductus arteriosus stenting (reverse Potts shunt procedure) and another patient underwent atrial septostomy and then atrial stenting.

Regarding gender differences, two male patients died. As such, mortality was significantly higher in males (*p* < 0.05). No other statistically significant differences were found between genders.

### 4.1. Case 1

A young male refugee was diagnosed with IPAH in Italy at the age of 11. He was symptomatic for shortness of breath and desaturation on mild exertion (NYHA class III), while the saturation on room air was 90%. Echocardiography displayed severely elevated RVSP (Figure 1).

Severely increased pressure on the right side of the heart was confirmed on RHC. An aggressive strategy with three drugs (sildenafil, bosentan, and iloprost) was chosen. Following a multidisciplinary meeting, the patient was put on the waiting list for lung transplantation, but died while waiting for a donor because of heart failure.

### 4.2. Case 2

A male patient was diagnosed with IPAH as an infant. He grew up with the disease and felt quite well with the help of double therapy (sildenafil and macitentan). A small patent of ductus arteriosus acted as a pop-off, thus providing some relief until he reached adolescence. Since then, his condition deteriorated quickly. Atrial septostomy followed by placement of a fenestrated device was decided as a palliative approach. Two years after the procedure the patient is in good clinical condition and still alive (Figure 2 and Figure 3).

### 4.3. Case 3

A female patient had IPAH since infancy. She was treated with sildenafil, bosentan, and epoprostenol over the years, with a good response. However, her clinical conditions worsened during adolescence. As such, the patient underwent a patent ductus arteriosus stenting (reverse Potts shunt) which was aimed at decompressing the pressure-overloaded right ventricle. It was at the expense of a certain degree of desaturation at the lower half of the body (saturation measured at forefinger was 93%, whereas at foot big toe was 82%). The patent is still alive 3 years following the procedure.

## 5. Discussion

PAH in paediatrics usually manifests in the form of IPAH, congenital heart disease–related PAH, and PPHN [12]. Notwithstanding the different PH aetiologies, they have in common analogous histopathologic pulmonary vasculature abnormalities. As such, the therapeutic approach is similar [13,14]. IPAH in children is rare (<10:1,000,000) and its incidence is 1–2:1,000,000/year [15]. Many worldwide registries exist including children with PAH of various aetiologies. However, even in these registries, the number of patients recorded with IPAH is limited to a few decades [16,17]. One of them is the American Registry to Evaluate Early and Long-Term PAH Disease Management (REVEAL) registry. It is a 55-centre, observational study including subjects less than 18 years with group I PAH. More than half of them are suffering from IPAH. There is a slight female predominance (60%) which is consistent with the findings in the presented case series [18]. IPAH is a progressive and lethal disease, as testified by the fact that 80% of the sample size was in NYHA functional class III or IV, and deserves an aggressive treatment provided in specialised centres. The NYHA class is one of the factors predicting survival, with class IV mean survival being less than 6 months [9]. Symptoms in paediatric IPAH include failure to thrive and growth impairment, most of all in children < 5 years, tachycardia, tachypnoea owing to cardiac insufficiency, and cyanosis in the presence of an atrial shunt [19]. According to the current guidelines, PH diagnosis is based on echocardiographic probability followed by RHC [20]. The latter is the “gold standard” in diagnosing and evaluating the degree and prognosis of IPAH. It is considered an undangerous technique when it is carried out at centres with expertise in the field (complication rates and mortality in adulthood are 5.7% and 0.2%, respectively) [21]. Nevertheless, as RHC may be linked with the occurrence of severe adverse events (in 1–3% of cases, mostly in infants and those in worse clinical conditions), it should be executed in paediatric PH centres with a high level of expertise. Risks and benefits should be balanced case by case [22,23]. On echocardiography, average right ventricular contractility, expressed by means of TAPSE, was reduced [24]. TAPSE varies with age and not having used z-scores is a limitation. However, their use is not widely diffused yet [25]. Prognosis in IPAH is strongly linked to right ventricular contractility [26]. Sato and Coll demonstrated that right ventricular systolic function in those with PH is better estimated by TAPSE than by right ventricular fractional change area [27]. With echocardiography, RVSP proved to be severely increased. However, estimated RVSP by Doppler can be significantly different compared with RHC in approximately half of the cases [28]. This is consistent with our findings. The RHC data confirmed that mPAP was severely raised as a consequence of a very elevated PVR, which tends to be progressive in IPAH [29]. The average mRAP was elevated as well, reflecting right ventricular overload. It is an established risk factor for mortality [30]. Despite having high right ventricular afterload, CI was quite preserved. The European Society of Cardiology (ESC) and European Respiratory Society (ERS) guidelines include CI and SVO_2_ among the haemodynamic parameters that are useful for risk stratification in IPAH patients. Not only that, but a recent study showed that SVO_2_ is able to predict long-term mortality better than CI [31]. In the studied case series, the average SVO_2_ was low–normal. Multi-drug therapy is currently suggested for those affected by PAH. This is the reason why the majority of our patients were taking three drugs, which were provided in sequence (e.g., phosphodiesterase inhibitor first, then endothelin receptor antagonist, and finally prostanoid). However, one of the patients in the case series was administered an aggressive upfront combination of three drugs, with an excellent clinical outcome. This is consistent with that previously reported in the literature regarding severe non-reversible PAH [32]. In two patients, a pop-off shunt was created artificially for palliation/bridge to transplant treatment. In the first of the two, a very small ductus arteriosus was stented (reverse Potts shunt) to open a wide interconnection between the descending portion of the aorta and the pulmonary artery resulting in a right-to-left shunt. Potts shunt decompresses the right ventricular chamber and secures enough cardiac output, causing hypoxemia in the lower limbs [33]. In the second patient, an intracardiac right-to-left shunt was created by an atrial septostomy. The artificially created atrial septal defect was then stented to maintain it open. In severe PAH, this palliative procedure decompresses the right ventricle, preserves its function, improves haemodynamics, and increases systemic blood flow [34]. About half of those in the study were subjected to lung transplantation or are waiting for that. IPAH is progressive, and for non-responders to medicines, lung transplantation is the last option [35]. This is not without any risks, because of the extremely high mortality in these patients, owing to the complexity of the treatment and the frequent onset of post-surgical insufficiency of the left ventricle. The other problems are increased mortality, impaired kidney and liver functions, and at times thrombocytopathy related to continuous prostaglandin infusion [36]. Overall, notwithstanding the progress in medical therapy, IPAH continues to represent a serious challenge, particularly in the paediatric population, with the need for lung transplantation and significant mortality in a relatively short time. However, metabolomics, a recent technique, could be of hope, which reveals the complex cascade of metabolic reactions underpinning a number of pathologies including PH/PAH [37]. In the meantime, due to the rarity of the disease, multicentre studies trying to enroll as many patients as possible are needed to shed light on its still obscure characteristics [38].

## Figures and Tables

**Figure 1 children-10-00518-f001:**
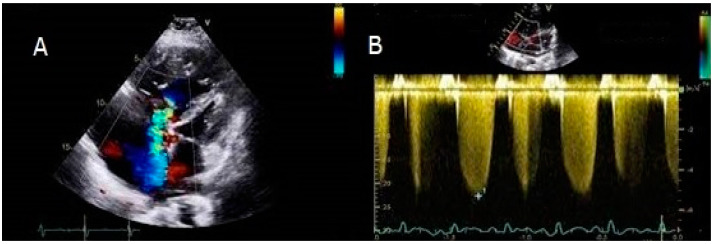
Apical four chambers view showing a very dilated right ventricle and severe tricuspid valve regurgitation. The interventricular septum bulged toward the left ventricle (panel **A**). The RVSP was significantly increased (panel **B**).

**Figure 2 children-10-00518-f002:**
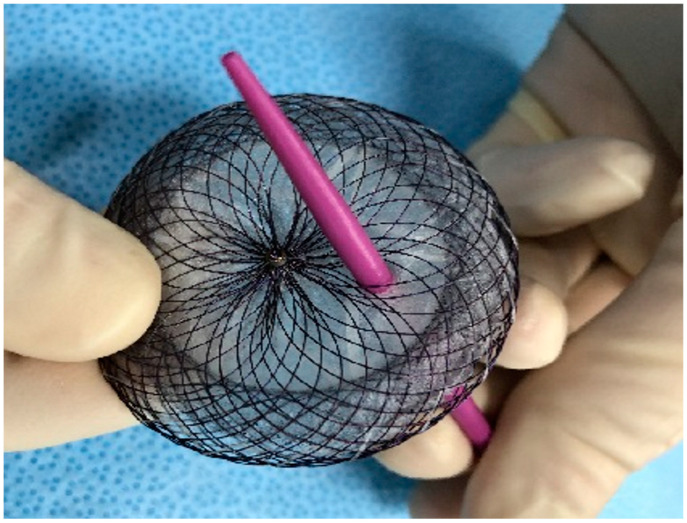
Fenestrated atrial septal defect.

**Figure 3 children-10-00518-f003:**
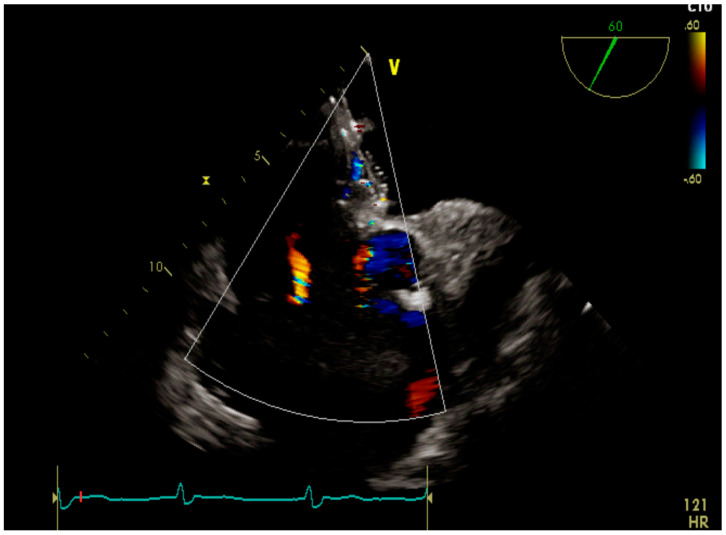
Transesophageal echocardiography showing the fenestrated device.

## Data Availability

Not applicable.

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
