# Peer review of "Idiopathic Pulmonary Arterial Hypertension in Paediatrics Represents Still a Serious Challenge: A Case Series Study"

_children, 2023, doi:10.3390/children10030518_

Round 1

Reviewer 1 Report

The authors provide a well described case serie study about idiopathic arterial hypertension in children and adolescents.

Please check the formula on line 101.

On echocardiographic assessment TAPSE is used to evaluate right ventricular function. As values vary with age, could you please provide an explanation why z-scores for TAPSE were not used?

Author Response

Dear Colleague,

Thanks for the time spent in reviewing our manuscript in an attempt to improve it.

In relation to the raised points:

  • In the formula at line 107 “x” has been omitted
  • We agree that TAPSE varies with age. However, the use of z-scores for TAPSE is not widely diffuse yet. This has been addressed in the Discussion paragraph. A new reference has been added.

All the revised parts are in red. Other changes have been made in accordance with that suggested by the other reviewers.

Reviewer 2 Report

Basaren et al. report a series of 10 children suffering from idiopathic pulmonary arterial hypertension. The data were collected during a 20 year time period. Their observations confirm IPAH to have a poor prognosis.

IPAH in children represents a rare disorder. Nonetheless, I believe that the present study has several flaws.

(1) Low number of patients. Several registry data are available including several hundreds of patients.

(2) Long term follow-up. The mean follow-up time was around 6.6 years - this is not really long term.

(3) Methods section. At least this is a description of well known echocardiography and right heart catheterization.

(4) Results. This is a quite short, abstract-like paragraph. It would have been of interest how the measured variables (PAP, RV-FAC, etc. ...) changed over time. The data are reported, but i cannot identify the point of time, when they were collected (at first presentation?, after 1-year?,...?)

(5) Statistics. It seems to be very difficult to do any statistical testing in a cohort with 10 patients. E.g. 6 females and 4 male is really significant out of 10 patients? The same is true for the statement, that mortality was significantly higher in male patients (nearby, I do not find how many males/females died).

(6) Discussion. The authors themselves cite large international multi center registries - which challenges the importance of this 10 patients series.

I understand that IPAH is a very rare and severe disease. But from my point of view, the present series more likely represents a personal story report than a scientific study. The grade of novelty seems to be quite low.

Author Response

Dear Colleague,

Thanks for the time spent in reviewing our manuscript with a view of improving it.

In relation to the raised points:

  • We agree that several registries exist with hundred of patients suffering from PAH of all aetiologies in paediatrics. However, even in these registries the number of patients with idiopathic pulmonary arterial hypertension is limited to a few decades. It has been added in the Discussion. The number of patients we have collected at two tertiary centres, over a period of twenty years, is just 10. They are the all the existing paediatric patients with the disease in Ireland (5 million of inhabitants) and in Naples (Italy) and surroundings (7 million of inhabitants). In our case series we have also showed some palliative options which are available in selected cases, e,g. atrial septostomy and reverse Potts shunt. It has been highlighted presenting three selected cases.
  • The adjective “long term” has been deleted (line 60) and replaced with “relatively short term” has been added (line 217)
  • Thanks for your appreciation about the accurate description
  • This section has been enlarged with the description of three selected cases. We do not have data regarding changes over time. As reported in the Table (for some reasons I get the impression that you do not have received it from the Journal) echo and RHC were measured at last follow-up. It has been reported in the text now (Line 122)
  • Two male patients died. Even with a certainly small sample size, some statistically significant differences were found, including RVSP on echo vs RHC
  • We agree with you. This is a small case series and not a registry.

All the revised parts are in red. Other changes have been made in accordance with that suggested by the other reviewers.

Reviewer 3 Report

This manuscript describes a case series in which 10 paediatric patients with pulmonary hypertension were enrolled, and clinical data were collected from a follow-up of average 6 years. the authors found that most of the patients relied on heavy intervention, and prominent mortality and transplantation needs still existed despite medical therapy. This study focuses on the outcome and long-term prognosis of pulmonary hypertension in children, which brings some scientific significance. I listed several major concerns need to be addressed. 

1.    Bring the table from the supplement to the main text to show more details.

2.    Please describe at least 3 typical cases individually from your observation, including baseline information, testing result, healthcare process and follow-up outcome.

3.    Show the typical images of your echocardiography test. Make a comparison of the images from the same patients during his/her follow-up if possible.

Author Response

Dear Colleague,

Thanks for the time spent in reviewing our manuscript with a view of improving it.

In relation to your suggestions:

  1. We agree that having Table 1 amid the paper would be much better. Unfortunately, it is impossible to bring the table from Supplementary material to the text, because of the rigid rules of the Journal regarding formatting tables. Just portrait tables are allowed in terms of layout, whereas Table 1 has a landscape format. It does not depend on us
  2. Three cases have been described according to your suggestion
  3. Some images have been added.

Round 2

Reviewer 3 Report

This manuscript describes a case series in which 10 pediatric patients with pulmonary hypertension were enrolled, and clinical data were collected from a follow-up of average 6 years. The authors found that most of the patients relied on heavy intervention, and prominent mortality and transplantation needs still existed despite medical therapy. This study focuses on the outcome and long-term prognosis of pulmonary hypertension in children, which brings some scientific significance. The authors responded to my questions very well and made thoughtful revisions to make this work more solid.  I listed several concerns need to be addressed. 

1.    Figure 1 legend, change the “wiev” to “view”.

2.    Line 161, change the “grow-up” to “grew up”. 

Author Response

Dear Colleague,

Many thanks for the further time spent in reviewing our paper.

The highlighted 2 typos have been amended now.